# DUAL-CYCLE CONSISTENCY LEARNING FOR WEAKLY SUPERVISED PHRASE GROUNDING

## ABSTRACT

Weakly supervised phrase grounding (WSPG) aims to localize objects referred by phrases without region-level annotations. The state-of-the-art methods use vision-language pre-trained (VLP) models to build pseudo labels. However, their low quality could result in the ineffectiveness of the subsequent learning. In this paper, we propose a novel WSPG framework, Dual-cycle Consistency Learning (DCL). Firstly, we propose a vision-modal cycle consistency to localize the referred objects and reconstruct the pseudo labels. To provide a conditional guidance, we propose a visual prompt engineering to generate marks for input images. To further avoid localizing randomly, we design a confidence-based regularization to filter out redundant information in image and pixel levels. Secondly, we propose a language-modal cycle consistency to correctly recognize the referred objects. To correct their positions, we provide phrase-related boxes as supervision for further learning. Extensive experiments on benchmark datasets show the effectiveness of DCL, as well as its excellent compatibility with various VLP models. The source code will be available at GitHub after double-blind phase.

## 1 INTRODUCTION

Weakly supervised phrase grounding (i.e., WSPG) localizes referred objects based on phrase queries without any box annotation. The WSPG task has the potential to benefit various downstream works, such as image captioning (Liu et al., 2022b; Shi et al., 2021; Li et al., 2024; Wang et al., 2024b), vision-language navigation (Barthel et al., 2019; Li et al., 2021b; Wu et al., 2022; Eftekhar et al., 2024), and visual question answering (Wu et al., 2023; Chen et al., 2023; Xiao et al., 2024; Peng et al., 2024; You et al., 2024). Earlier works have employed the outputs of object detectors matched with phrases (Ren et al., 2015; Datta et al., 2019; Gupta et al., 2020; Wang et al., 2021; Wang & Specia, 2019; Rohrbach et al., 2016; Chen et al., 2018; Liu et al., 2021), or devised auxiliary tasks to offer effective supervisory information for the grounding network (Fang et al., 2015; Xiao et al., 2017; Javed et al., 2018; Zhang et al., 2018; Akbari et al., 2019; Arbelle et al., 2021). However, these approaches are suboptimal as they are constrained by their cross-modal alignment capabilities.

Recently, various WSPG methods leverage vision language pre-training (VLP) models to aid in grounding the target object. They rely on the attention maps of the VLP models as pseudo labels for training. These attention maps provide visual highlights of objects' locations in the images, and thus can be used to guide the optimization process. Previous VLP-based WSPG studies have developed two types: VLP-based methods with fine-tuning and those with parameters frozen. Fine-tuned VLP approach (He et al., 2023; Zeng et al., 2024) focuses on the localization by reducing the difference of pseudo labels during fine-tuning. Frozen VLP methods (Shaharabany et al., 2022; Shaharabany & Wolf, 2023; Gomel et al., 2023; Lin et al., 2024a) extract pseudo labels with VLPs and devise additional networks to refine the coarse pseudo labels. However, previous works disregard the low quality of pseudo labels. It could result in the ineffectiveness of subsequent learning.

Here, we divide the problems caused by low-quality pseudo labels into three categories: incompleteness, redundancy, and misrecognition (in Figure 1). **Firstly**, pseudo labels offer limited information, as they tend to convey category-level details without comprehensive positional context. As shown in the left example, the red highlight of *pretty lady* is salient but does not cover all necessary information. A naive idea of generating a similar highlight could overlook the object's localization information. Therefore, it becomes imperative to utilize the pseudo label as a starting point and

| Incompleteness | Redundancy | Misrecognition |
|---|---|---|
| 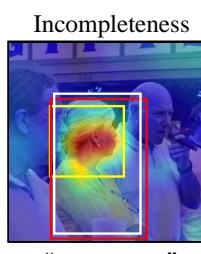 | 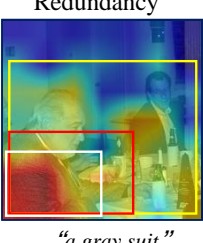 | 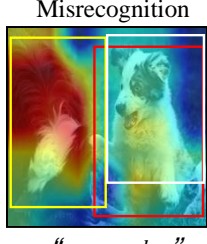 |
| *"a pretty lady"* | *"a gray suit"* | *"a puppy dog"* |

Figure 1: **Three challenging problems in VLP-based WSPG.** Incompleteness: grounding will focus on a portion of the target object. Redundancy: pseudo labels sometimes provide redundant information. Misrecognition: phrase-irrelevant objects are located. We illustrate WSPG's results (yellow box), ground-truth (red box), our results (white box), and pseudo labels (attention maps).

gradually obtain comprehensive information of the referred object during the reconstruction of the pseudo label. **Secondly**, VLP-based methods are sensitive to redundant information in pseudo labels. As shown in the middle example, highlight regions show *gray suit*, where the dimmer regions represent redundant information, and the brighter ones contain valuable information for learning. To accurately predict the localization of *suit*, it is essential to decouple the association between redundant information and the phrase *gray suit* during the training phase. In short, we need to refine the weakly supervised learning process by minimizing the harmful effects of redundant information present in pseudo labels. **Finally**, there is a wrong recognition for the object referred by the query phrase. As shown in the right example, if the red highlight encompasses an area of the *giant dog* instead of *puppy dog*, the model tends to erroneously recognize the instance *giant dog* based on the phrase *puppy dog*. To overcome this ambiguity, we need to design a language consistency strategy that reduces referential confusion and ensures precise positional supervision, thereby accurately localizing the referred object.

To mitigate the negative effects of low-quality pseudo labels, we propose a novel WSPG framework, Dual-cycle Consistency Learning (i.e., DCL). **Firstly**, we introduce a vision-modal cycle consistency to prevent incompleteness and redundancy. It learns to localize the referred object and reconstruct pseudo labels. Specifically, we employ a grounding network and a recovery module to perform two consecutive grounding operations. We use the pseudo label as a prompt to identify and ground the referred object, and subsequently align the initial pseudo label with the second-time grounding result. In order for the pseudo labels to provide category-level details, we treat the pseudo labels as the conditional guidance of the network. We also develop a visual prompt engineering, which equips input images with mark prompts. Furthermore, we utilize pseudo labels to provide constraints during the first-time grounding process. It could avoid our grounding network localizing randomly. In order to filter out redundant information from pseudo labels, we design a regularization method that imposes image-level and pixel-level confidence constraints. **Secondly**, we propose a language-modal cycle consistency to address the correspondence ambiguity between the localized object and the query phrase. This approach represents concepts and details in a caption format, and recognizes whether the localized objects are the referred ones by distinguishing between captions and query phrases. To correct the location based on the phrases, we propose a region captioning verification process to generate caption-box pairs for prospective locations. Subsequently, we select optimal boxes from them for further consistency learning.

To sum up, the main contributions of our work are three-fold.

- We propose a novel VLP-based WSPG framework to mitigate the adverse effects of low-quality pseudo labels. To the best of our knowledge, we are the first to explore the detrimental impact of VLPs' pseudo labels in WSPG and to propose an effective strategy.

- We design a dual-cycle consistency learning for WSPG. A vision-modal cycle consistency aims to augment the functionality of pseudo labels. A language-modal cycle consistency aims to recognize and correct the referred object based on the query phrase.

- We conduct extensive experiments on three benchmark datasets to verify the effectiveness of our framework and its excellent compatibility with different VLP models.

## 2 RELATED WORK

**Vision Language Pre-trained Models.** Pre-trained models have significantly advanced the domain of CV and NLP by learning from large datasets (Kenton & Toutanova, 2019; He et al., 2019). This trend has prompted research on handling both visual and textual data, known as Vision-Language Pre-trained (VLP) Models (Li et al., 2019; Chen et al., 2020; Tan & Bansal, 2019). For instance, CLIP (Radford et al., 2021) has demonstrated superior performance in aligning images with their corresponding texts through pre-training on extensive internet-sourced image-text pairs. Other outstanding works include TCL (Yang et al., 2022) and ALBEF (Li et al., 2021a). Additionally, some applications have incorporated VLP models into generative frameworks (Rombach et al., 2022; Chefer et al., 2023). These works enable textual descriptions to be represented as features for image generation. A recent surge has employed pre-trained models for grounding-related tasks (Subramanian et al., 2022; Shtedritski et al., 2023; Liu et al., 2024; Yang et al., 2024; Wang et al., 2024c). However, the potential negative implications of these models have received limited attention.

**VLP-based WSPG.** VLP models have been increasingly employed for WSPG. This task focuses on localizing objects within images based on query phrases, not relying on any region-level annotation. Most recent methods involve either fine-tuning (He et al., 2024; Zeng et al., 2024) or maintaining a frozen state of VLP models (Shaharabany & Wolf, 2023; Lin et al., 2024a). The former methods adjust VLP models to better localize objects by reducing inconsistencies in pseudo labels over multiple fine-tuning phases. In contrast, the latter methods do not alter the pre-trained VLP models but instead extract attention heatmaps. These heatmaps are used as pseudo labels to train an independent WSPG network. Following the pioneering work (Shaharabany et al., 2022), subsequent efforts (Gomel et al., 2023; Lin et al., 2024b) have further refined the model's localization through collaborative learning with visual subtasks, such as segmentation and detection. However, the low quality of pseudo labels could result in the ineffectiveness of the subsequent model's learning.

**Consistency Learning for Grounding.** Our approach to WSPG can be regarded as a consistency learning. Language related consistency learning has been explored using classical weakly supervised referring expression grounding (REG) (Liu et al., 2019; 2022a; Zhang et al., 2023; Wang et al., 2024a; Liu et al., 2021). For vision consistency learning, Zhu et al. (2017) pioneering proposed unpaired translation for image generation. Recently, Cyco (Wang et al., 2024a) proposes a grounding captioning consistency method. In this method, a collaborate learning network is designed for REG and image captioning. To train the network, the data including the image, the text description, and the bounding box are required. However, Cyco ignores the cost associated with manually labeling the bounding box. In addition, Cyco does ignore the problem of incorrect pseudo labels which could harm the model's performance. In this paper, we propose a WSPG framework using VLP models. We design a dual-cycle consistency learning to mitigate the negative effects of pseudo labels.

## 3 METHODOLOGY

### 3.1 OVERVIEW

Given an image $I$ and a query phrase $T$, the task of phrase grounding requires the model to produce a bounding box $B$. To this end, a heatmap $H$ is generated as a helper. In VLP-based WSPG, the model is trained with image-phrase pairs and a pseudo label $A$ extracted with VLP models.

The overview of our proposed framework is shown in Figure 2. Our grounding network consists of an image encoder $\mathcal{E}_{img}(\cdot)$, a text encoder $\mathcal{E}_{txt}(\cdot)$, and a grounding decoder $\mathcal{D}_{gnd}(\cdot)$. The image encoder employs the last layer's output of the pre-trained CNN in ImageNet as visual embedding. The text encoder uses the text embedding branch of CLIP (VIT-B/32), which is frozen. The grounding decoder only consists of two up-sampling layers. It firstly fuses bi-modal features, and converts high-dimensional fusion features into grounding heatmaps $H$. The feature fusion calculates the similarity between text features and visual ones, $A_M = \mathcal{E}_{img}(I) \otimes \mathcal{E}_{txt}(T)$. The attention is then given as $R_M = \mathcal{E}_{img}(I) \circ A_M$, in which the symbol $\circ$ means Hadamard product.

To mitigate the detrimental effects of low-quality pseudo labels, we propose a dual-cycle consistency learning (DCL) framework, including vision-modal cycle consistency and language-modal cycle consistency. The former takes pseudo labels as prompts, enabling the grounding network to learn to

Figure 2: Overview of our VLP-based WSPG framework. Two types of heatmap transition are based on vision-modal and language-modal cycle consistency learning.

localize during the reconstruction of pseudo labels. The latter recognizes the referred object based on its corresponding phrases and corrects its position. Subsequently, we describe our DCL in details.

## 3.2 VISION-MODAL CYCLE CONSISTENCY

We devise a novel approach that leverages pseudo-label reconstruction to localize referred objects. Our method involves a two-stage grounding process. In the grounding network, we use the pseudo label $A$ as the prompt to ground the object referred by the phrase $T$. This produces the first grounding heatmap $H$. The process is formulated as follows,

$$H = \mathcal{D}_{gnd}\left(\mathcal{E}_{img}(\mathcal{P}_{img}(I, A)), \mathcal{E}_{txt}(T)\right) \tag{1}$$

where $\mathcal{P}_{img}(I, A)$ denotes the prompt function for an image with a pseudo label. Subsequently, in the recovery module, we once again use the grounding heatmap $H$ as the prompt to ground the referred object. This produces the second grounding heatmap $H_R$. The formulate is given as follows,

$$H_R = \mathcal{D}_{gnd}\left(\mathcal{E}_{img}(\mathcal{P}_{img}(I, H)), \mathcal{E}_{txt}(T)\right) \tag{2}$$

The recovery module has the same structure as the grounding network. To enhance the similarity between the grounding heatmap $H_R$ and the pseudo label $A$, we propose the visual consistency loss. We use the mean squared error (MSE) criterion, i.e.,

$$L_{VI} = \frac{1}{n} \sum_{n=1}^{N} \left((H_R)_n - A_n\right)^2 \tag{3}$$

We refer to this scheme as vision-modal cycle consistency.

**Conditional Visual Prompt Engineering.** The heatmaps similar to pseudo labels for capturing the region of referred objects are previously proposed (Shaharabany et al., 2022; Gomel et al., 2023; Lin et al., 2024b). However, these heatmaps largely contain the salient information, struggling to delineate details. Additionally, these methods solely relying on a phrase could fail to accurately convey the intended grounding content. To provide not only supervision but also category-level details, the pseudo labels are treated as the conditional guidance. Thus, we employ visual prompt engineering by marking regions on the input image, thereby providing a conditional guidance. Specifically, to highlight each referred object in the input images, we utilize six variants of prompt engineering, including *Keypoint*, *Red Circle* (Shtedritski et al., 2023), *Red Box* (Chen et al., 2020), *Mask*, *Crop* (Yao et al., 2021), and *Image Blur* (Yang et al., 2024). Then the input image can be generated

as $P_{img}(I, A)$, where $P_{img}$ contains six approaches of visual prompt engineering. Note that *Image Blur* is controlled by the standard deviation in Gaussian blur kernel $\delta$.

In the absence of constraints, the grounding network tends to localize objects randomly. A naive idea is to adopt pseudo labels to constrain the first-stage grounding results $H$. However, it inevitably suffers from redundant information. Thus, we need to design a confidence-based regularization method to remove redundant information in pseudo labels.

**Confidence-based Regularization.** To reduce the interference of redundant information in pseudo labels, we design a confidence-based regularization method. The regularization involves image-level confidence (IC) and pixel-level confidence (PC). In IC, each pseudo label's highlighted area reflects the confidence of the label's quality. In PC, each pixel in the pseudo label indicates the confidence of the position corresponding to the query phrase. Thus, we attempt to ignore those untrusted labels and positions. This is shown in Figure 3. Specifically, given the pseudo label $A \in R^{H \times W}$ of an image, we exact its bounding box as $B(A) \in R^{H_B \times W_B}$. The image-level and pixel-level confidence maps are obtained as follows,

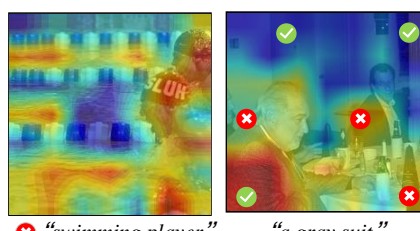

"swimming player"    "a gray suit"

Figure 3: Two types of redundant information can be filtered by confidence-based regularization.

$$IC(A) = \frac{H_B \times W_B}{H \times W} \quad \text{and} \quad PC(\alpha, \beta) = \max(\max(A(\alpha, \beta)), 1 - \max(A(\alpha, \beta))) \quad (4)$$

where $\alpha$ and $\beta$ means the pixels of pseudo labels or grounding heatmaps. We convert MSE to a confidence-based version as follows,

$$L_{CM} = \begin{cases} \frac{1}{N} \sum_{n=1}^{N} (H_n - A_n)^2 & , IC \leq \mu \,\&\, PC \geq \gamma \\ 0 & , IC > \mu \text{ or } PC < \gamma \end{cases} \quad (5)$$

The hyper-parameters $\gamma$ and $\mu$ help the grounding network in ignoring pixels and pseudo labels with low confidence. Similarly, we propose a dice loss (Li et al., 2020) $L_{CD}$ based on the confidence to measure similarities. The formula is given as follows,

$$L_{CD} = \begin{cases} 1 - 2 \times \frac{\sum_{n=1}^{N} (H_n \cdot A_n)}{\sum_{n=1}^{N} H_n^2 + \sum_{n=1}^{N} A_n^2} & , IC \leq \mu \,\&\, PC \geq \gamma \\ 0 & , IC > \mu \text{ or } PC < \gamma \end{cases} \quad (6)$$

Note that we set the confidence-based loss to 0 if the confidence score is out of the range given by $\mu$ and $\gamma$. In addition, we re-normalize the non-zero loss values within a batch.

## 3.3 LANGUAGE-MODAL CYCLE CONSISTENCY

We employ a captioning approach to represent the objects' concepts and details within that region. This caption is then compared to the query phrase to ensure the language consistency. Specifically, we generate a bounding box $B(H)$ based on the grounding heatmap $H$. We then use the caption module (Li et al., 2022) to describe the content of the boxed region as $T_B$. While there may exist semantic similarities between the caption and the query phrase, discrepancies in content can arise. For example, "*image of wide and blue air*" and "*image of this is the sky*", these samples are difficult to be recognized. To this end, we introduce a regularization using CLIP text encoder to extract embeddings, which facilitates the evaluation of semantic similarity. To make sure that the grounded region contains the referred object, we introduce $L_{DE}$ to minimize the difference between the embeddings of caption $T_B$ and query phrase $T$, while maximizing the difference between $T_B$ and a negative sample $T_N$. $L_{DE}$ is defined as follows,

$$L_{DE} = 1 - CLIP_{txt}(T_B, T) + CLIP_{txt}(T_B, T_N) \quad (7)$$

where $CLIP_{txt}$ denotes the score calculated solely by the CLIP text encoder. Note that we treat the description "*image of colorful patches*" as a negative sample. This setting is based on its commonness in captions generated by the caption module. Such captions typically arise when the grounded region has either incomplete or ambiguous instances.

Less object-independent information in captions assists in judging the relevance of the localized object to the referred object. Thus, we adopt spaCy (Subramanian et al., 2022) for Name Entity

Recognition (NER) in phrases. To align the primary object within the grounded region and the subject of the query phrase, we compute the cosine score between the second recognized nouns in $T$ and $T_B$. For example, "*image of a train pulling carts*" vs. "*image of this is the train*") are used. The similarity loss $L_{SU}$ is defined as follows,

$$L_{SU} = 1 - \cos\left(\mathcal{E}_{txt}(NER(T_B)), \mathcal{E}_{txt}(NER(T))\right) \tag{8}$$

While the language-modal cycle consistency is effective for recognizing incorrect referred objects, the guidance on how to correct to the appropriate phrase-related position is still lacking. Thus, we give the network the guidance of phrase-related box as additional position supervision.

**Boxes Generation and Selection.** We design a region captioning verification process to generate the corresponding box annotations for potential objects. To identify regions likely to contain instances, we adopt several techniques, including selective search algorithm (Uijlings et al., 2013), bounding box generation algorithm (Shaharabany et al., 2022), and random proposals. Thus, we generate proposals $\{b_1, ..., b_n\}$. A challenge is to discern the specific concepts and details of these instances. To this end, we employ the caption module (Li et al., 2022) to generate caption expressions $\{t_1, ..., t_n\}$ for proposed regions. We filter out semantically repetitive proposals. The semantic redundancy may manifest as varied descriptions of the same object, such as "*black coat*" versus "*padded jacket*", or "*red bike*" versus "*small bicycle*". Specifically, we use CLIP text encoder (Radford et al., 2021) to translate the explicit captions into latent features. Based on the latent features, we build clusters and ensure that the data points within each cluster exhibit uniformity in the feature space. Within each cluster, instances are ranked based on scores calculated using the similarity to the mean feature representation. We then select the top-$k$ scoring instances, aiming to filter out instances that lack the semantic coherence with the cluster. Finally, we select the cluster whose semantic similarity is closest to the phrase embedding $\mathcal{E}_{txt}(T)$, as $Z = \{(t_k, b_k)\}_{k=1}^{K}$.

**Position Consistency Learning.** To provide reliable positional annotations, we use CLIP text encoder $\mathcal{E}_{txt}$. It calculates the text similarity score between the query phrase $T$ and the captions in cluster $Z$. The box $b_i$ associated with the top-1 most similar caption $t_i$ is then propagated. Similarly, cross-modal similarity scores between the given image and captions are also calculated to derive a box $b_j$, using the complete CLIP. The propagation processes, as shown in Figure 4, are shown as:

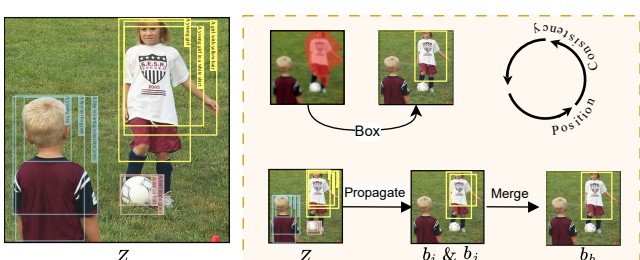

Figure 4: The results of region captioning verification process (left). The process of consistency learning (right).

$$\arg\max_{Z_i} = S_{text}\left(Z_i = (t_i, b_i) | T, Z\right) \quad \text{and} \quad \arg\max_{Z_j} = S_{cross}\left(Z_j = (t_j, b_j) | T, Z\right) \tag{9}$$

where $Z$ represents the cluster selected by the region captioning verification process. We merge these two obtained boxes $b_i$ and $b_j$ to form $b_h$, representing the smallest box enclosing both. To refine the grounding result's boundaries, we utilize $L_{BOX}$ (Gomel et al., 2023) and $L_{GIOU}$ (Rezatofighi et al., 2019) as follows,

$$L_{BOX} = \|B(H) - b_h\|_1 \quad \text{and} \quad L_{GIOU} = 1 - \left(\frac{|B(H) \cap b_h|}{|B(H) \cup b_h|} - \frac{|c_h \setminus (B(H) \cup b_h)|}{|c_h|}\right) \tag{10}$$

where $c_h$ is the smallest box containing $B(H)$ and $b_h$. We set the position loss as $L_{PO} = L_{BOX} + L_{GIOU}$. In addition, we set language-modal consistency loss as $L_{LC} = L_{DE} + \tau L_{SU} + \epsilon L_{PO}$.

Thus, we summarize the total loss for our model as follows,

$$L_{Total} = L_{VI} + \lambda_1 L_{CM} + \lambda_2 L_{CD} + \lambda_3 L_{LC} \tag{11}$$

In the inference phase, we feed the prompted image $P_{img}(I, A)$ and query phrase $T$ to the grounding network. The module then generates a grounding heatmap $H$. Finally, we adopt the bounding box generation method proposed by Shaharabany et al. (2022), obtaining the bounding box.

## 4 EXPERIMENTS

### 4.1 DATASETS

Four datasets are used in our experiments. **Flickr30K Entities** (Plummer et al., 2015) contains 224K phrases describing bounding boxes in 31K images, and each image includes five captions. We also select 1000 images from the test split to evaluate as used in MG (Akbari et al., 2019). **ReferIt** has 20,000 images and 99,535 segmented regions in IAPR TC-12 (Grubinger et al., 2006) and SAIAPR-12 (Chen et al., 2017) datasets, respectively. There exist approximately 130K entity captions. We used the same 9K training, 1K validation, and 10K test datasets as in MG (Akbari et al., 2019). **MSCOCO 2014** (Lin et al., 2014) contains 82,783 train images and 40,504 validation images. Each image is described with 5 captions. The training split in MG is used. **Visual Genome** (Krishna et al., 2017) consists of 77,398 training images, 5,000 test images, and 5,000 validation images. Each image possesses a series of annotations which are in a free-text format.

### 4.2 BASELINES AND METRICS

We chose typical VLP models as our **backbones**. 1) Classical image-text matching models, i.e., CLIP (Radford et al., 2021), ALBEF (Li et al., 2021a) and TCL (Yang et al., 2022). 2) Text-to-image generation models, i.e., Stable Diffusion (Rombach et al., 2022) and Attend-and-Excite (Chefer et al., 2023). In addition, we used two typical WSPG methods and seven VLP-based WSPG methods as **baselines**. 1) Classical WSPG baselines, i.e., MG (Akbari et al., 2019) and Gbs (Arbelle et al., 2021). 2) VLP-based WSPG includes g (Shaharabany et al., 2022), g++ (Shaharabany, 2023), BBR (Gomel et al., 2023), SelfEQ (He et al., 2024), TAS (Lin et al., 2024a), VPT (Lin et al., 2024b) and APR (Zeng et al., 2024).

Two **metrics**, i.e., "pointing game" accuracy (Akbari et al., 2019) and bounding box accuracy (Shaharabany et al., 2022) are used. "Pointing Game" accuracy measure the percentage of predicted maximum points of the heatmap that lie within the bounding box ground-truth. Bounding Box accuracy measure the percentage of heatmap bounding boxes that have an IoU greater than 0.5 for the testing set of "image-query" pairs.

### 4.3 IMPLEMENTATION DETAILS

For a fair comparison, we used VGG-16 as the image encoder in our framework. For VLP models, all pseudo labels are extracted with the interpretable method GAE (Chefer et al., 2021). Note that the specific backbone layers that GAE acts on are different. For CLIP, we use all layers of the visual encoder for GAE. For ALBEF and TCL, we use the third layer of the cross-modality encoder. For Stable Diffusion and Attend-and-Excited, we use the second layer and fifth layer of the last cross-attention block. To ensure that multiple losses belong to the same scale, the weights in our loss function were set as follows: $\lambda_1 = 16, \lambda_2 = 4, \lambda_3 = 0.5, \tau = 4$, and $\epsilon = 10$.

### 4.4 QUANTITATIVE RESULTS

We conduct experiments using the same training and inference processes with MG (Akbari et al., 2019). In subsequent analysis, our framework uses *Image Blur* (Yang et al., 2024) as the only conditional visual prompt engineering.

**Comparison with SoTA Methods.** We compare our method with other WSPG methods on Visual Genome (VG), Flickr30k Entities, and ReferIt. We distinguish VLP-based WSPG methods with similar pseudo labels from three sources, including CLIP (Radford et al., 2021), ALBEF (Li et al., 2021a), and g (Shaharabany et al., 2022). For a fair comparison, we combine DCL with three types of pseudo labels. Differ from the first two types of pseudo labels, we use g's output heatmaps as pseudo labels. The experimental results are shown in Table 1. It shows that our framework exceeds the previous state-of-the-art methods in all settings. Our approach works for different forms of the training data (i.e., MS-COCO and VG) and the testing data (i.e., Flickr30K Entities, VG, and ReferIt). In addition, our method could alleviate the impact of low pseudo-label quality.

**Compatibility with VLP Models**. We report experimental results under different VLP models, including TCL (Yang et al., 2022), CLIP (Radford et al., 2021), and ALBEF (Li et al., 2021a).

| Model | VG Trained | | | | | | MS-COCO Trained | | | | | |
|---|---|---|---|---|---|---|---|---|---|---|---|---|
| | Point Accuracy | | | Bbox Accuracy | | | Point Accuracy | | | Bbox Accuracy | | |
| | *VG* | *Flickr* | *ReferIt* | *VG* | *Flickr* | *ReferIt* | *VG* | *Flickr* | *ReferIt* | *VG* | *Flickr* | *ReferIt* |
| MG | 48.76 | 60.08 | 60.01 | 14.45 | 27.78 | 18.85 | 47.94 | 61.66 | 47.52 | 15.77 | 27.06 | 15.15 |
| Gbs | 53.40 | 70.48 | 59.44 | - | - | - | 52.00 | 72.60 | 56.10 | - | - | - |
| g | 62.31 | 75.63 | 65.95 | 27.26 | 36.35 | 32.25 | 59.09 | 75.43 | 61.03 | 27.22 | 35.75 | 30.08 |
| APR | 60.43 | 78.07 | 63.75 | - | - | - | - | - | - | - | - | - |
| **Ours (CLIP)** | **64.26** | **78.54** | **68.95** | **29.61** | **39.85** | **35.07** | **61.81** | **77.74** | **62.27** | **27.94** | **40.51** | **31.33** |
| SelfEQ | - | 81.90 | 67.40 | - | - | - | - | 84.07 | 62.75 | - | - | - |
| **Ours (ALBEF)** | **62.82** | **82.12** | **68.01** | **28.43** | **39.95** | **30.65** | **60.16** | **84.46** | **63.62** | **26.35** | **37.66** | **28.63** |
| g++ | 66.63 | 79.95 | 70.25 | 30.95 | 45.56 | 38.74 | 62.96 | 78.10 | 61.53 | 29.14 | 46.62 | 32.43 |
| BBR | 63.51 | 78.32 | 67.33 | 31.02 | 42.40 | 35.56 | 60.05 | 77.19 | 63.48 | 28.77 | 47.26 | 30.63 |
| TAS | 58.07 | 76.69 | 70.86 | 27.31 | 45.63 | 35.70 | 60.31 | 77.85 | 62.63 | 29.58 | 45.46 | 33.41 |
| VPT | 62.72 | 80.03 | 68.21 | 27.40 | 45.60 | 34.76 | 60.74 | 81.15 | 64.14 | 27.65 | 45.09 | 31.14 |
| **Ours (g)** | **67.17** | **81.54** | **70.93** | **32.64** | **45.80** | **39.69** | **63.21** | **82.65** | **64.38** | **30.04** | **47.88** | **33.58** |

Table 1: Performance of WSPG methods on the test splits. The best results are shown in boldface.

| Method | Training | Test Point Accuracy | | | Test Bbox Accuracy | | |
|---|---|---|---|---|---|---|---|
| | | *VG* | *Flickr* | *ReferIt* | *VG* | *Flickr* | *ReferIt* |
| TCL | - | 55.36 | 79.95 | 54.29 | 22.04 | 32.14 | 20.86 |
| TCL+ours | VG | 65.88($\uparrow$10.52) | 82.79($\uparrow$2.84) | 64.55($\uparrow$10.26) | 30.96($\uparrow$8.92) | 41.27($\uparrow$9.13) | 36.80($\uparrow$15.94) |
| TCL+ours | MS-COCO | 63.06($\uparrow$7.70) | 82.96($\uparrow$3.01) | 62.24($\uparrow$7.95) | 31.14($\uparrow$9.10) | 44.69($\uparrow$12.55) | 33.35($\uparrow$12.49) |
| CLIP | - | 54.72 | 72.47 | 56.76 | 16.70 | 25.56 | 19.10 |
| CLIP+ours | VG | 64.26($\uparrow$9.54) | 78.54($\uparrow$6.07) | 68.95($\uparrow$12.19) | 29.61($\uparrow$12.91) | 39.85($\uparrow$14.29) | 35.07($\uparrow$15.97) |
| CLIP+ours | MS-COCO | 61.81($\uparrow$7.09) | 77.74($\uparrow$5.27) | 62.27($\uparrow$5.51) | 27.94($\uparrow$11.24) | 40.51($\uparrow$14.95) | 31.33($\uparrow$12.23) |
| ALBEF | - | 51.59 | 78.15 | 57.41 | 20.25 | 28.30 | 15.79 |
| ALBEF+ours | VG | 62.82($\uparrow$11.23) | 82.12($\uparrow$3.97) | 68.01($\uparrow$10.60) | 28.43($\uparrow$8.18) | 39.95($\uparrow$11.65) | 30.65($\uparrow$14.86) |
| ALBEF+ours | MS-COCO | 60.16($\uparrow$8.57) | 84.46($\uparrow$6.31) | 63.62($\uparrow$6.21) | 26.35($\uparrow$6.10) | 37.66($\uparrow$9.36) | 28.63($\uparrow$12.84) |
| Stable Diffusion+ours | VG | 55.31 | 65.41 | 53.06 | 18.88 | 28.65 | 20.11 |
| Stable Diffusion+ours | MS-COCO | 52.89 | 63.96 | 54.22 | 19.06 | 30.41 | 20.73 |
| Attend-and-Excite+ours | VG | 57.83 | 68.80 | 54.76 | 19.92 | 30.07 | 22.53 |
| Attend-and-Excite+ours | MS-COCO | 59.36 | 68.92 | 53.33 | 18.58 | 32.25 | 21.09 |

Table 2: The results using different VLP models and generative models in our method. For a fair comparison, all pseudo labels are extracted by the identical method (Chefer et al., 2021).

The experiments are shown in Table 2. It shows that our DCL is effective across a spectrum of VLP models. Note that our DCL achieves a superior grounding performance in comparison to the VLP models. Furthermore, we also report the results using Stable Diffusion (Rombach et al., 2022) and Attend-and-Excite (Chefer et al., 2023) in the last four rows. These generative models produce results based solely on phrases, without using images in the MSCOCO and VG datasets. The synthetic images along with pseudo labels extracted via GAE (Chefer et al., 2021) are combined with the original phrases to constitute the training corpus for our DCL. This scheme does not employ visual prompts during the inference stage, as generative models are incapable of generating attention heatmaps relevant to input images. The results show unsatisfactory performance when using the generative model as the backbone of our approach. This suboptimal performance could be attributed to two factors. 1) The inaccurate images generated by generative models can lead to cumulative errors in the model's learning. 2) Generative models' propensity to generate object-centered outputs contrasts with the complex backgrounds of input images (Plummer et al., 2015; Krishna et al., 2017; Chen et al., 2017). It leads to distributional discrepancies when outputs used as the training data. However, the grounding performance achieved through this approach serves as an indicator of the generative model's capability in capturing the semantics of the given phrase. Attend-and-Excite (Chefer et al., 2023) exhibits a superiority in generating images that convey the semantics of the query phrase. In contrast, the other model produces less favorable results.

## 4.5 ABLATION STUDY

In this section, we empirically investigate how the performance of our framework is affected by different model settings. All models were trained on VG (Krishna et al., 2017), and we used pseudo labels extracted from CLIP (Radford et al., 2021).

| M | Prompt | VI | CM | CD | LC | Test Point Accuracy | | | Test Bbox Accuracy | | |
|---|--------|----|----|----|----|----|----|----|----|----|----|
| | | | | | | VG | Flickr | ReferIt | VG | Flickr | ReferIt |
| ✓ | | | | | | 48.42 | 57.85 | 50.11 | 13.47 | 16.10 | 17.92 |
| | ✓ | | | | | 54.49(↑6.07) | 68.72(↑10.87) | 56.88(↑6.77) | 16.87(↑3.40) | 24.10(↑8.00) | 22.14(↑4.22) |
| | ✓ | ✓ | | | | 57.63(↑3.14) | 72.44(↑4.06) | 60.94(↑4.06) | 17.55(↑0.68) | 25.46(↑1.46) | 24.02(↑1.88) |
| | ✓ | ✓ | ✓ | | | 60.27(↑2.64) | 75.85(↑3.41) | 64.23(↑3.29) | 26.11(↑8.56) | 35.53(↑10.07) | 30.31(↑6.29) |
| | ✓ | ✓ | | ✓ | | 60.49(↑2.86) | 75.63(↑3.19) | 64.32(↑3.38) | 23.58(↑6.03) | 33.38(↑7.92) | 27.85(↑3.83) |
| | ✓ | ✓ | ✓ | ✓ | | 62.94(↑5.31) | 76.61(↑4.17) | 67.19(↑6.25) | 26.34(↑8.79) | 36.35(↑10.89) | 31.18(↑7.16) |
| | ✓ | ✓ | ✓ | ✓ | ✓ | **64.26**(↑1.32) | **78.54**(↑1.93) | **68.95**(↑1.76) | **29.61**(↑3.27) | **39.85**(↑3.50) | **35.07**(↑3.89) |

Table 3: The ablation results of various components. "M" represents our baseline. "Prompt" denotes the conditional visual prompt engineering. "VI" means the vision-modal cycle consistency. "CM" and "CD" represent confidence-based losses. "LC" means the language-modal cycle consistency.

**Model Components.** We explore the performance of DCL with various components. **Firstly**, we construct a simple network as our grounding network. The network only adopts the MSE loss from g (Shaharabany et al., 2022) as its training objective. This network serves as a baseline for subsequent comparison. In addition, five key components are involved. The experimental results are presented in Table 3. The using of five components in our framework

| $L_{DE}$ | $L_{SU}$ | $L_{PO}$ | Test Point Accuracy | | | Test Bbox Accuracy | | |
|----------|----------|----------|-----|--------|---------|-----|--------|---------|
| | | | VG | Flickr | ReferIt | VG | Flickr | ReferIt |
| ✓ | | | 63.07 | 77.07 | 67.56 | 28.03 | 38.16 | 33.28 |
| | ✓ | | 63.61 | 77.64 | 68.12 | 27.47 | 37.60 | 32.23 |
| | | ✓ | 64.02 | 78.21 | 68.69 | 28.59 | 38.71 | 34.34 |
| ✓ | ✓ | | 63.76 | 77.94 | 67.22 | 28.04 | 38.25 | 33.49 |
| | ✓ | ✓ | 64.00 | 78.24 | 68.67 | 28.84 | 38.83 | 34.14 |
| ✓ | | ✓ | 63.97 | 78.48 | 68.73 | 29.59 | 39.60 | 34.93 |

Table 4: The ablation results of three losses in the language-modal cycle consistency.

consistently enhances the performance. We observe that a better performance could be achieved with conditional visual prompt engineering. It corroborates the efficacy of our enhancements over the original method. Then adding the vision-modal cycle consistency strategy can boost the performance. The result demonstrates that the effectiveness and compatibility of two-stage grounding process. In addition, confidence-based regularization contributes most to the performance gain. We suppose that our method filters out noisy pseudo labels while tries to remove visual noise from the pseudo labels. The language-modal cycle consistency strategy also demonstrates an improvement in the model's performance. This verifies that our approach can mitigate the influence of the error accumulation during training. **Secondly**, we investigate the effectiveness of different losses in our DCL. The results are shown in Table 4. We observe that the ablation strategy's performance is lower than the our complete strategy. The three losses help localized objects follow the semantics of phrases, and ensure that the grounded region contains the targeted object.

**Hyperparameters.** We conduct experiments on hyperparameter, $\mu$ and $\gamma$ in confidence-based losses. The results are shown in Table 5. The optimal values for MSE are 0.95 and 0.95. For dice loss, these two values are 0.95 and 0.99. Figure 5 presents our DCL's grounding performance when varying the hyperparameters, $k$ and $\delta$. We observe that our framework achieves the best performance when the hyperparameter $k$ is set to 5. A higher or lower values could weaken the quality of positional annotations, which are selected by the region captioning verification process. We also ablate the standard deviation of the Gaussian blur kernel in *Image Blur*. The deviation of 100 achieves the best performance.

| CM | | CD | | Test Point Accuracy | | | Test Bbox Accuracy | | |
|----|---|----|---|-----|--------|---------|-----|--------|---------|
| $\mu$ | $\gamma$ | $\mu$ | $\gamma$ | VG | Flickr | ReferIt | VG | Flickr | ReferIt |
| 1.00 | 1.00 | 1.00 | 1.00 | 62.57 | 76.64 | 67.20 | 26.85 | 37.60 | 32.85 |
| 0.95 | 1.00 | 1.00 | 1.00 | 63.28 | 77.14 | 67.65 | 27.68 | 38.47 | 33.95 |
| 0.95 | 0.95 | 1.00 | 1.00 | 63.51 | 77.62 | 67.72 | 29.35 | 39.01 | 34.47 |
| 0.95 | 0.95 | 0.95 | 1.00 | 64.03 | 78.17 | 68.30 | 28.72 | 39.36 | 34.89 |
| 0.95 | 0.95 | 0.95 | 0.95 | 63.11 | 77.38 | 67.52 | 29.04 | 39.42 | 34.97 |
| 0.90 | 0.95 | 0.95 | 0.99 | 63.68 | 77.83 | 68.33 | 28.76 | 38.73 | 34.08 |
| 0.95 | 0.90 | 0.95 | 0.99 | 62.98 | 76.98 | 67.58 | 27.32 | 37.86 | 33.32 |
| 0.95 | 0.95 | 0.90 | 0.99 | 63.97 | 78.22 | 68.60 | 29.43 | 39.64 | 34.88 |
| 0.95 | 0.95 | 0.95 | 0.90 | 63.01 | 77.02 | 67.62 | 28.89 | 38.91 | 34.24 |
| 0.95 | 0.95 | 0.95 | 0.99 | **64.26** | **78.54** | **68.95** | **29.61** | **39.85** | **35.07** |

Table 5: Ablations of image-level and pixel-level confidences. "CM" and "CD" represent confidence-based MSE and dice loss.

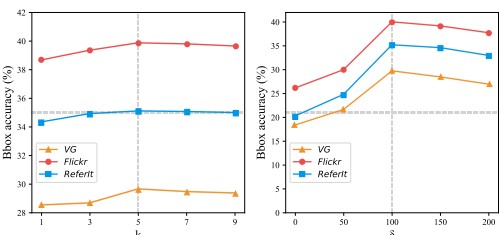

Figure 5: The performance with parameters $k$ and $\delta$ in our DCL are shown, respectively. The results are conducted on three datasets.

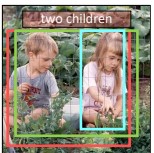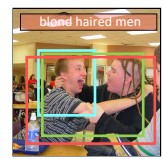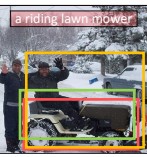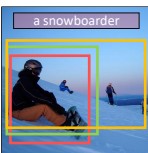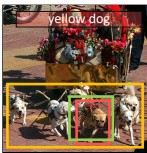

Figure 6: The visualization of grounding results of six testing examples. The red boxes are ground-truth. The green boxes are generated by our best model. The blue boxes are generated by our model without the region captioning verification process. In addition, the orange boxes are produced by our model without the conditional visual prompt engineering.

## 4.6 QUALITATIVE ANALYSIS

We show the qualitative results from the Flickr30K Entitiess in Figure 6. In the left three examples, the key factor to localizing the referred object is leveraging the positional annotations from the region captioning verification process. In the absence of the region captioning verification process, there is a deviation in the estimates of *silver car*, *two children* and *bland haired men* compared to the ground-truths. In the right three examples, we observe that the model trained with the prompt localizes target objects much better than the one trained without the prompt component. In the absence of conditional guidance, the positioning of *mover, snowboarder, and dog* tends to be larger than expected. We conclude that both the proposed approaches play an essential role in accurately grounding the referred objects.

## 4.7 LIMITATIONS

Our network has limited performance on domain-specific data, such as remote sense and industrial abnormal datasets. A few results are shown in Figure 7. The first example fails because our method can only select a rough range and cannot locate each target object. The second example fails since our positioning had redundant parts. This phenomenon is attributed to the fact that commonly used VLP models are unable to establish strong cross-modal associations for these domains, resulting in inaccurate positioning. We will introduce more data to enhance the generality of our framework. In addition, the current DCL paradigm is designed for static imagery and requires significant advancements to adapt to dynamic video streams, such as continuous updates of refining matching concepts over time, correction of erroneous hypotheses, and robust tracking mechanisms for regions.

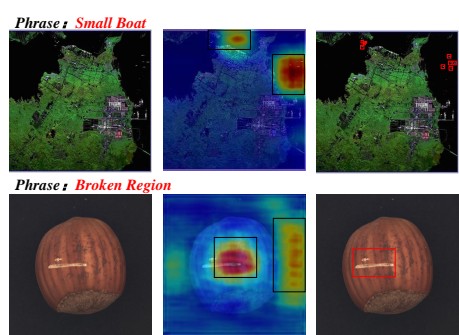

Figure 7: Failure cases of our method. The middle column represents our results.

## 5 CONCLUSION

In this paper, we propose a novel framework, Dual-cycle Consistency Learning (DCL) for WSPG. We propose a vision-modal cycle consistency to learn to ground the referred objects in the process of reconstructing the pseudo labels. This consistency prevents incompleteness and redundancy problems. We also propose a language-modal cycle consistency to learn to recognize the referred objects and correct their positions. This consistency mitigates the misrecognition problem based on the given phrase. Extensive experiments on benchmark datasets show that our framework achieves state-of-the-art performance and has excellent compatibility with different VLP models. In the future, we will study the application of our framework to related multimodal tasks, such as vision-language navigation and visual question answering.

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

## A    BROADER IMPACTS

Our research introduces a novel weakly supervised phrase grounding paradigm that improves phrase grounding performance, facilitating the development of multimodal interaction systems and benefiting people's daily lives. Furthermore, we explore weakly-supervised training, saving human efforts in data annotation. Our framework is validated on large-scale public vision-language datasets and does not leverage noise in the data, ensuring fairness and unbiasedness in the grounding results. In contrast, the failure of this technique may lead to an inaccurate multimodal understanding and cause the mistake of the system based on the grounding results.

## B    BASELINES

a. Selected VLP models: We introduce typical models from image-text matching (CLIP (Radford et al., 2021), ALBEF (Li et al., 2021a) and TCL (Yang et al., 2022)) and text-to-image generation (Stable Diffusion (Rombach et al., 2022), Attend-and-Excite (Chefer et al., 2023)). 1) CLIP is a joint vision and language model pre-trained using over 400 million images I and their corresponding captions T. It is comprised with two networks, image Encoder and text Encoder. The pre-training process of CLIP utilizes contrastive learning, which maximizes the cosine similarity between cross-modal pairs and minimizes the score between different images and captions. 2) ALBEF composes of a text decoder, an image encoder, and a multimodal fusion encoder. It relies on three widely used objectives for visual and textual representation learning: image-text matching, masked language modeling and a contrastive loss. 3) TCL, a two-stream model, is an enhanced version of ALBEF, which introduces three contrasting modules: Cross-modal Alignment (CMA), Intramodal Contrastive (IMC), and Local Mutual Information Maximization (LMI). These modules are designed to maximize the mutual information between matching images and texts and maximize global mutual information. 4) Stable Diffusion operates in the latent space of an autoencoder. First, an encoder E is trained to map a given image into a spatial latent code. A decoder is then tasked with reconstructing the input image. Given the trained autoencoder, a denoising diffusion probabilistic model (DDPM) operates over the learned latent space to produce a denoised version of an input latent at each timestep. During the denoising process, the diffusion model can be conditioned on an additional input vector. In Stable Diffusion, this additional input is typically a text encoding produced by a pre-trained CLIP text encoder. 5) Attend-and-Excite is an enhanced version of Stable Diffusion, which uses an attention-based formulation and guides the diffusion model to refine the cross-attention units to attend to all subject tokens in the text prompt.

b. Compared baselines: MG (Akbari et al., 2019), Gbs (Arbelle et al., 2021), g (Shaharabany et al., 2022), g++ (Shaharabany, 2023), BBR (Gomel et al., 2023), SelfEQ (He et al., 2024), TAS (Lin et al., 2024a), VPT (Lin et al., 2024b) and APR (Zeng et al., 2024). 1) MG maximizes the likelihood that a caption word appears in a distribution. It exploits multiple levels of feature maps of a DCNN, as well as word and sentence embeddings extracted from a character-based language model. The model is guided by a multi-level multi-modal attention mechanism which outputs activated visual features in each level. 2) Gbs uses the source separation technique to ground the phrase to the image pixels. The insight is to synthesize text-to-image regions by random alpha-blending of arbitrary image pairs. The query phrase is used as condition for a non-hybrid query image. 3) g utilizes the interpretable heatmap from CLIP as the supervision. In order to provide pixel-level supervision, the network utilizes CLIP to distinguish between the foreground and background of the output heatmap. 4) g++ designs a self-supervised segmentation training method to further optimise the grounding network. This method gives good results by optimising the grounding annotation alone without changing the loss function of g. 5) BBR proposes a self-supervised object detection method for joint learning with the grounding network. 6) SelfEQ helps the grounding network to recognise uncommon phrases by distillation, while this method pre-processes grounding-related phrase data with the assistance of LLM. 7) TAS proposes a triple alignment strategy for solving the zero-shot phrase grounding under weak supervision. 8) VPT proposes a visual prompt tuning method to effectively alleviate the local optimal problem of WSPG network. 9) APR constructs attribute, relation and priority grounding benchmarks to evaluate the compositional reasoning on grounding tasks for different models.

## C  PROMPT VARIANTS

There are six types of prompt engineering variants used in our framework: 1) **Red Box** (Chen et al., 2020) serves as a visual prompting. It generates red boxes as markers on images. The position of *Red Box* is the same as that of the bounding box. 2) **Keypoint** entails placing a small circle at the center of *Red Box*. 3) **Red Circle** (Shtedritski et al., 2023) corresponds to an inscribed ellipse derived from *Red Box*. 4) **Mask** serves as a form of prompting by masking the region within input image corresponding to the highlight region. 5) **Crop** (Yao et al., 2021) serves as a form of prompting by cropping the image region along *Red Box*. 6) **Image Blur** (Yang et al., 2024) serves as a form of prompting by blurring the region within input image. This region corresponds to the highlight region. *Image Blur* is controlled by the standard deviation in the Gaussian blur kernel.

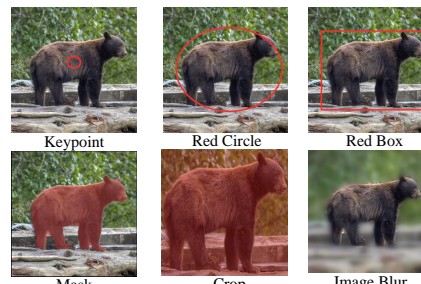

Figure 8: Six variants of visual prompt for the query phrase "*brown bear*".

## D  OPTIMIZATION OF OTHER WSPG METHODS

Our DCL can easily incorporate other WSPG methods into its own framework, in visual explanation algorithms (Chefer et al., 2021; Zhou et al., 2021; Subramanian et al., 2022), and the state-of-the-art WSPG models (Shaharabany et al., 2022; Lin et al., 2024b). To summarize, we treated these methods as pseudo-label generators, and formed several two-stage weakly supervised grounding baselines. Table 6 shows the performance comparison of these baselines, with the results obtained using their official codes. All baselines have notable performance improvement for grounding results. In addition, we also report the performance of our conditional visual prompt engineering combined with g (Shaharabany et al., 2022) and

| Method | Backbone | Flickr | Setting | Flickr |
|---|---|---|---|---|
| GAE | CLIP | 25.56 | +*DCL* | 39.85(↑14.29) |
| MaskCLIP | CLIP | 34.26 | +*DCL* | 41.01(↑6.75) |
| GradCAM | CLIP | 23.18 | +*DCL* | 38.37(↑6.75) |
| g | CLIP + VGG | 36.35 | +*prompt*
+*DCL* | 38.17(↑1.82)
45.80(↑9.45) |
| VPT | CLIP + VGG | 45.60 | +*prompt*
+*DCL* | 45.66(↑0.06)
46.23(↑0.63) |

Table 6: Comparison with SoTA WSPG methods evaluated using the bounding box accuracy. All models were trained on Visual Genome dataset.

VPT (Lin et al., 2024b). "+prompt" represents that we utilize the *Image Blur* method on their input images. The results show that prompt engineering has a positive impact on the weakly supervised learning process. It also shows that our method could optimize other grounding methods, and has good compatibility.

## E  EFFECTIVENESS OF DIFFERENT PROPOSALS

Another important factor is the quality of proposals, which are generated based on region captioning verification process. We therefore investigated the effect of using different proposals. These proposals are extracted from different bounding box generation methods: selective search algorithm (Uijlings et al., 2013), pseudo label's bounding box (Chefer et al., 2021) and random proposals. As shown in Ta-

| $S_{text}$ | $S_{cross}$ | Test Point Accuracy | | | Test Bbox Accuracy | | |
|---|---|---|---|---|---|---|---|
| | | VG | Flickr | ReferIt | VG | Flickr | ReferIt |
| ✓ | | 63.94 | 77.87 | 68.50 | 29.43 | 39.06 | 34.51 |
| | ✓ | 61.72 | 76.65 | 66.19 | 28.38 | 36.87 | 33.62 |
| ✓ | ✓ | **64.26** | **78.54** | **68.95** | **29.61** | **39.85** | **35.07** |

Table 7: Performance of our network with different positional annotations.

ble 8, increasing the variety of proposals can improve the performance of our framework. The multiple proposal generation algorithms give the process of annotating more options. In addition, we ablated the method of obtaining positional annotations as shown in Table 7. Two variants reduce performance across all metrics in datasets. Mixing $S_{text}$ and $S_{cross}$ schemes attains the best result.

| SS | Ran | GAE | Test Point Accuracy | | | Test Bbox Accuracy | | |
|---|---|---|---|---|---|---|---|---|
| | | | VG | Flickr | ReferIt | VG | Flickr | ReferIt |
| ✓ | | | 62.69 | 74.85 | 66.85 | 26.76 | 38.17 | 32.14 |
| | ✓ | | 60.50 | 73.95 | 64.91 | 23.76 | 34.30 | 30.18 |
| | | ✓ | 62.23 | 76.28 | 66.79 | 28.61 | 38.53 | 33.90 |
| ✓ | ✓ | | 63.01 | 75.76 | 67.02 | 27.30 | 38.26 | 33.00 |
| | ✓ | ✓ | 62.21 | 76.31 | 66.83 | 28.66 | 38.62 | 33.92 |
| ✓ | | ✓ | 63.55 | 77.96 | 68.28 | 29.10 | 39.18 | 34.45 |
| ✓ | ✓ | ✓ | **64.26** | **78.54** | **68.95** | **29.61** | **39.85** | **35.07** |

Table 8: Performance of our network with different proposals. *"Ran"* represents random proposals. We set its number as three.

| $\lambda_1$ | $\lambda_2$ | $\lambda_3$ | $\tau$ | $\epsilon$ | Test Point Accuracy | | | Test Bbox Accuracy | | |
|---|---|---|---|---|---|---|---|---|---|---|
| | | | | | VG | Flickr | ReferIt | VG | Flickr | ReferIt |
| 16 | 4 | 0.5 | 4 | 10 | **64.26** | **78.54** | **68.95** | **29.61** | **39.85** | **35.07** |
| 1 | 4 | 0.5 | 4 | 10 | 63.51 | 77.62 | 68.16 | 27.62 | 37.60 | 33.06 |
| 16 | 1 | 0.5 | 4 | 10 | 63.37 | 77.46 | 68.03 | 28.58 | 38.47 | 33.86 |
| 16 | 4 | 1 | 4 | 10 | 63.39 | 77.60 | 68.14 | 28.60 | 38.49 | 33.94 |
| 16 | 4 | 0.1 | 4 | 10 | 62.81 | 78.02 | 67.94 | 28.79 | 38.69 | 34.13 |
| 16 | 4 | 0.5 | 1 | 10 | 64.02 | 78.25 | 68.70 | 29.52 | 39.74 | 34.97 |
| 16 | 4 | 0.5 | 4 | 1 | 64.22 | 78.38 | 68.70 | 28.47 | 38.35 | 33.75 |

Table 9: The ablation results of various weight of hyper-parameters. The first row represents the settings for best performance.

## F  EFFECTIVENESS OF DIFFERENT VISUAL PROMPTS

We compare the sensitivity of DCL to different visual prompt variants. In this setting, visual prompts, as illustrated in Sec. C, were generated according to our proposed framework. Consequently, *Image Blur* shows superior performance demonstrated in Table 10. The application of "Bokeh" blurring serves to obfuscate the background while accentuating the object, thereby providing a clearer indication of its distinctive position within the scene. Additionally, this method facilitates the network's comprehension of the object's relationship with its surrounding context.

## G  MORE VISUALIZATIONS

In this section, we present the visualizations of our DCL's results for the weakly supervised phrase grounding task, as shown in Figure 9. The query phrases are displayed in the lower-left corner of the displayed images. The results reflect the alignment between instances and query phrases within the figure. The same cluster of caption-box pairs is indicated using identical colors, and all proposals and positional annotations are generated in the region captioning verification process.

## H  ADDITIONAL FAILURE CASES

In this section, we present additional failure case of our framework. As shown in Fig. 10, "a blue coat" belongs to "a reporter" but not "a new crew", but we ground "blue coats" instance of all people in the image. This is because our framework extracts only noun phrases without considering phrases in-context during the inference, leading to an inaccurate evaluation of the referred object's localization.

## I  LOSS WEIGHT ABLATION

In this section, we ablate the weights of loss items in Table 9. The first row represents the settings for best performance, and we present the hyper parameters in the Sec.4.3.

## J  ADDITIONAL TRAINING DETAILS

All models are trained on a GeForce A6000 Nvidia GPU. We use an SGD optimizer (batch size of 32 and an initial learning rate of 0.0003). We also set the optimizer momentum as 0.9 and weight decay as 0.0001. In addition, we use a random horizontal flip with 0.5 probability. Our network is optimized for 120 epochs, where pseudo labels are generated by CLIP (Radford et al., 2021), ALBEF (Li et al., 2021a), TCL (Yang et al., 2022) and g (Shaharabany et al., 2022). To save the training resource, we train our network without $L_{DE}$ and $L_{SU}$ for 115 epochs, and add both losses in the last five epochs. When we extract pseudo labels from stable diffusion (Rombach et al., 2022) and Attend-and-Excited (Chefer et al., 2023), our network is optimized for 1 epoch due to the time-consuming generation of images.

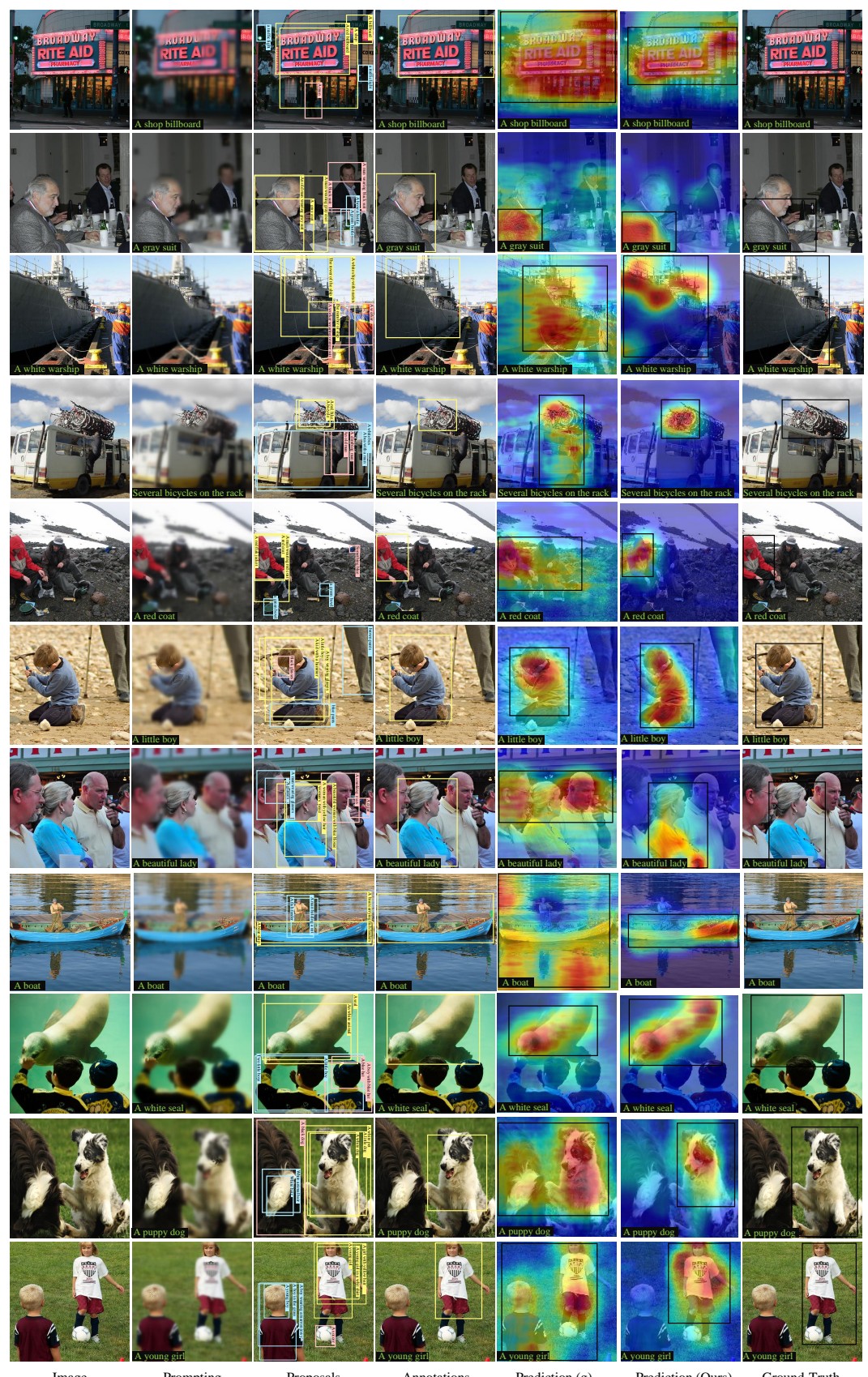

Figure 9: Visualization of DCL results on the phrase grounding task under the Flickr30K Entities, VG, and ReferIt datasets.

| Method | Test Point Accuracy | | | Test Bbox Accuracy | | |
|---|---|---|---|---|---|---|
| | VG | Flickr | ReferIt | VG | Flickr | ReferIt |
| Red Circle | 62.26 | 76.54 | 66.95 | 27.61 | 37.85 | 33.07 |
| Keypoint | 59.03 | 72.14 | 61.06 | 18.40 | 26.23 | 20.28 |
| Red Box | 63.92 | 77.87 | 68.83 | 28.63 | 38.80 | 34.64 |
| Mask | 60.97 | 76.10 | 64.17 | 19.38 | 27.46 | 20.51 |
| Crop | 62.81 | 76.95 | 67.23 | 21.61 | 30.00 | 24.77 |
| Image Blur | **64.26** | **78.54** | **68.95** | **29.61** | **39.85** | **35.07** |

Table 10: Performance of our network with different visual prompt engineering variants.

*Caption : A new crew and a reporter in a blue coat make a film in the rain.*

*Image*     *Our DCL Result*     *Ground Truth*

Figure 10: Failure cases of our method. Row #1 presents the query phrase and the sentence. Column #2 presents the failure case for grounding entities in context. Column #3 presents ground-truth.

| Method | Backbone | CUDA Memory | Training Time | Inference Time | IPS | Acc |
|---|---|---|---|---|---|---|
| AdaptingCLIP | CLIP | 3289 MB | - | 22.67 min | 0.74 | 23.18 |
| MaskCLIP | CLIP | 2004 MB | - | 1.02 min | 16.39 | 34.26 |
| GAE | CLIP | 4324 MB | - | 2.28 min | 7.30 | 25.56 |
| g | CLIP + VGG | 19364 MB | 2900 min | 1.90 min | 8.77 | 36.35 |
| VPT | CLIP + VGG | 19364 MB | 2930 min | 1.90 min | 8.77 | 45.60 |
| DCL* (ours) | CLIP + VGG + BLIP | 18954 MB | 86400 min | 4.18 min | 3.99 | 39.85 |
| DCL† (ours) | g + VGG + BLIP | 18222MB | 5121 min | 4.18 min | 3.99 | 45.80 |
| | CLIP + VGG + BLIP | 18906 MB | 5155 min | 4.57 min | 3.65 | 39.85 |
| | ALBEF + VGG + BLIP | 20095 MB | 5205 min | 5.14 min | 3.24 | 39.95 |
| | Stable Diffusion + VGG + BLIP | 32397 MB | 25239 min | 1.90 min | 8.77 | 28.65 |

Table 11: Comparison of training and inference cost. IPS: Image per GPU second. ∗ denotes that extracting positional annotation is realized during the model training phase. † indicates that extracting box annotation is implemented prior to model training.

## K  INFERENCE SPEED AND COMPUTATION

In this section, we present the computation and inference speed of our network in different settings, as shown in Table 11. All trainable models were trained on Visual Genome and achieved inference on Flickr30K Entities, gaining their bounding box accuracy.

