# OpenReview forum: "Dual-cycle Consistency Learning for Weakly Supervised Phrase Grounding"
_ICLR.cc/2025/Conference — ICLR 2025 Conference Withdrawn Submission_

### Official Review · Reviewer_UgBx · 2024-10-30

**Soundness:** 2
**Presentation:** 3
**Contribution:** 2
**Rating:** 3
**Confidence:** 4

**Summary:**

This paper proposes a Weakly Supervised Phrase Grounding (WSPG) framework, named Dual-cycle Consistency Learning (DCL). DCL utilizes vision-modal cycle consistency to localize objects and language-modal cycle consistency to correctly recognize objects. DCL utilizes various prompt engineering techniques to generate visual prompts based on pseudo labels. The effectiveness of the proposed method is demonstrated through comparison experiments and ablation studies.

**Strengths:**

1.	The paper is well-organized, and the expression is clear, making it easy to follow.
2.	The figures in the paper are clear, making it easy to understand the framework.
3.	The related work section of this paper is comprehensive.

**Weaknesses:**

1.	About the training and inference cost. Table 11 in the supplementary material shows that DCL requires nearly twice the time of VPT to achieve comparable performance to VPT. For example, in the setting of “g + VGG + BLIP”, the IPS (image per GPU second) of DCL is 3.99 while the IPS of VPT is 8.77. Would it be possible to reduce the training and inference cost of DCL further?
2.	About the image-level confidence. In Equation 4, the authors use the proportion of the pseudo label’s area as the image-level confidence. It would be better if the authors could provide a detailed explanation to clarify the reason for this approach.
3.	About the vision-model cycle consistency. In the vision-model cycle, the authors utilize pseudo labels A as the constraints of HR. However, since pseudo label A contains noise (e.g., redundant information), how to ensure the quality of the generated HR?
4.	About the experiment settings. In Table 11 of the supplementary material, what does the "Backbone" column represent? For example, what does the meaning of "Stable diffusion + VGG + BLIP" setting? Additionally, this paper lacks a detailed analysis of performance under different Backbone settings.
5.	About the recovery module. In Equations (1) and (2), the authors use Dgnd to denote both the grounding network and the recovery module. Are these two networks sharing weights, i.e. are they the same network? Additionally, why is it necessary to enhance the similarity between Hr and the pseudo labels A instead of directly using H?

**Questions:**

My questions are as described in the weaknesses section.

---

### Official Review · Reviewer_MRKd · 2024-11-03

**Soundness:** 2
**Presentation:** 1
**Contribution:** 2
**Rating:** 3
**Confidence:** 4

**Summary:**

This paper studies the problem of weakly supervised phrase grounding, which aims to localize objects referred by phrases without region-level annotations. As the auto-generated pseudo labels are usually of low quality, the authors propose a dual-cycle consistency learning (DCL) approach. In the proposed approach, the vision-modal cycle consistency is designed to localize the referred objects, whereas the language-modal cycle consistency is designed to recognize the referred objects. Experiments on multiple benchmark datasets show the effectiveness of the proposed approach.

**Strengths:**

1. The problem of low quality pseudo labels (incompleteness, redundancy, and misrecongition) in weakly supervised phrase grounding is well motivated.
2. The overall idea of two cycle consistencies seems reasonable.

**Weaknesses:**

The major weaknesses of this paper are its poor (terrible) writting quality and less convincing experimental results.

1. The proposed algorithms look ad-hoc and lack solid technical contributions. While the high-level idea of the two cycle consistencies seems reasonable, it is unclear why and how the proposed algorithm can help address the low quality issues of auto-generated pseudo labels. More specifically, although three quality issues (incompleteness, redundancy, and misrecongition) of pseudo labels are listed in the introduction section, it's hard to see the connection how they are addressed by the proposed algorithm. After reading the introduction and the method sections twice, I still find it difficult to understand the algorithm intuition.

2. The method section needs significant writting improvement. It includes many details without high level intuition explained. The mathematical notations are terrible, often used without explanation, making it hard to follow. To list a few:
a) L158: What are the dimensionalities of $\mathcal{E}\_{\text{img}}(I)$ and $\mathcal{E}\_{\text{txt}}(T)$? They have almost the same notations. But $\mathcal{E}\_{\text{img}}(I)$ seems to keep the image shape $W \times H$, whereas $\mathcal{E}\_{\text{txt}}(T)$ is just a vector.
b) L194: How is the prompt function $\mathcal{P}\_{\text{img}}(I,A)$ defined and implemented?
c) L203: In Eq. (3), what does $n$ (and $N$) mean? Is it the index for pixels or for training images?
d) L232: exact --> extract. How to extract a bounding box $B(A)$ from the psedudo label $A$?
e) L236: Eq. (4) is totally messed up. What does it mean to compute the max of a single value as in $\text{max}(A(\alpha, \beta))$? And what exactly are the meanings of $\alpha$ and $\beta$? Are they pixel coordinates, or pixel values at a given position? As this equation is messed up and looks very ad-hoc, it's hard to understand this confidence-based regularization and vision-modal cycle consistency.
f) L240: I understand that $IC$ is a scalar. Is $PC$ also a scalar? Where are $\alpha$ and $\beta$ gone as in Eq. (3)?
g) In both Eq. (5) and Eq. (6), $n$ and $N$ are used without definition.
h) L262 and L266: How do you find negative samples $T_N$? It seems that you only treat "*image of colorful patches*" as a negative sample? If this is true, it is too ad-hoc and of little help to contrastive learning.
i) L271: Primary object identified as "the second recognized noun" is too ad-hoc and may be wrong for many cases.
j)  L292: "we select the cluster whose semantic similarity is closest to ..." From this description, it seems that you only select one cluster, but you ended up getting $K$ clusters.
k) L307: Eq. (9) is hard to follow, as $S_{text}$ and $S_{score}$ are not explained. Without being able to understand this equation, it's hard to understand language-modal cycle consistency.
l) L366: As image blur is used in the experiments, to make the paper self-explained, please briefly descibe how it is computed.

3. While the experimental results seem having better results than previous works, this reviewer found that for the ALBEF-based methods the paper missed a result from [1], also cited in the paper as APR (Zeng et al., 2024). APR [1] reported better results based on ALBEF than this paper:
ALBEF (point accuracy): 75.04 (VG), 84.49 (Flickr), 69.26 (ReferIt)
[1] Investigating compositional challenges in vision-language models for visual grounding, CVPR 2024

**Questions:**

1. The writting of this paper needs significant improvement. There are many grammatical errors, unclear sentences, and mathmatical notations in the paper. The current writting quality is far below the acceptance bar of ICLR.

2. The proposed algorithm looks ad-hoc, lacking clear intuition how to address the low quality issues of pseudo labels. As a result, three low quality issues are listed in the introduction section, but without being explicitly addressed.

3. The experimental results miss important baselines, making the results questionable.

4. As a general comment, the purpose of weakly supervised phrase grounding is to be able to leverage larger scale of training data. The paper should have a comparison with SOTA supervised phrase grounding algorithms and discuss whether it is possible, and under what kind of conditions, e.g. using more training data, that weakly supervised phrase grounding can outperform supervised algorithms.

---

### Official Review · Reviewer_NUmi · 2024-11-03

**Soundness:** 3
**Presentation:** 2
**Contribution:** 2
**Rating:** 5
**Confidence:** 4

**Summary:**

This paper argues the low-quality of pseudo labels in weakly supervised phrase grounding methods using the VLP pre-trained models. To solve the problem, the paper proposed a dual-cycle consistency learning framework. Three types of low-quality pseudo labels are categorized, incompleteness, redundancy and misrecognition.  The vision-modal cycle consistency is proposed to prevent incompleteness and redundancy by localizing the referred object and reconstructing the pseudo labels.

**Strengths:**

-	The method seems correct
-	The writing and organization seem clear
-	SOTA results are achieved and good ablation studies

**Weaknesses:**

-	Although with the good analysis of three error types, the major issue is the lack of theoretical insights or validated experimental proofs.
-	We don’t understand the statistics off these error types and how exactly the proposed modules tackle them. The effectiveness of the evaluations need more statistical validations
-	The novelty consists of integration of several engineering techniques. The idea of consistency learning is commonly used to reconstruct the pseudo labels or validate the pseudo labels. The exact reason of why the proposed method works need more investigations.

**Questions:**

see weakness

---

### Official Review · Reviewer_VMCv · 2024-11-03

**Soundness:** 2
**Presentation:** 2
**Contribution:** 2
**Rating:** 5
**Confidence:** 4

**Summary:**

This paper addresses the challenge of Weakly Supervised Phrase Grounding (WSPG) by introducing a novel framework called Dual-cycle Consistency Learning (DCL). DCL enhances pseudo label reconstruction through vision-modal cycle consistency and employs visual prompt engineering for improved guidance. Additionally, it integrates language-modal cycle consistency to accurately identify referred objects and strengthens localization with phrase-related supervision. Experimental results demonstrate DCL's effectiveness and its compatibility with various vision-language pre-trained models.

**Strengths:**

1. The proposed Dual-cycle Consistency Learning strategy is interesting.

2. Extensive experiments validate its effectiveness and compatibility.

3. Ablation studies highlight the contributions of various modules, while qualitative results further illustrate their functionality.

**Weaknesses:**

1. The writing is difficult to follow and requires further refinement.

2. Figure 2 is also confusing, particularly in illustrating the overall pipeline and the relationships among different modules.

3. The paper states that pseudo labels serve as conditional guidance, providing not only supervision but also category-level details. What specifically do these category-level details refer to? Could the authors offer additional explanation or analysis?

4. How does the vision-modal cycle consistency address potential incompleteness?

5. In Lines 232–233, how is the bounding box of the pseudo label extracted? How does the highlighted area of the pseudo label indicate the confidence in the label's quality, especially considering the presence of small objects to be grounded?

6. Can the method handle phrases that include spatial descriptions, such as "the girl on the right"? The language-modal cycle consistency mechanism seems to overlook this aspect.

7. What are the trainable parameters and model size of the proposed method? How does this compare to previous approaches?

**Questions:**

See Weakness.

---

### Official Review · Reviewer_G5SS · 2024-11-04

**Soundness:** 2
**Presentation:** 2
**Contribution:** 2
**Rating:** 5
**Confidence:** 5

**Summary:**

In this paper, the author propose a novel framework, Dual-cycle Consistency Learning (DCL) for WSPG. They propose a vision-modal cycle consistency to learn to ground the referred objects in the process of reconstructing the pseudo labels. The consistency prevents incompleteness and redundancy problems.

**Strengths:**

- The structure is complete.

**Weaknesses:**

Q1. The definition of "pseudo labels"  in this paper is somewhat strange. According to the existing literature, pseudo labels should refer to text labels.

Q2. There are many VLP-based grounding works that are not cited and discussed, such as CLIP-VG [1], CLIPREC [2], RefCLIP [3], etc.. Similarly, there are many weakly supervised grounding efforts that are not cited and discussed, just such as QueryMatch [4], PPT [5], etc.

Q3. The "double-loop consistent learning" proposed in this paper is a bit far-fetched, and it does not implement cycle consistency to some extent, which is quite different from the traditional concept of consistency learning such as CyCo.

Q4.Prompt has currently been widely used, and the prompt engineering proposed in this paper is not innovative.

--

[1] CLIP-VG: Self-paced Curriculum Adapting of CLIP for Visual Grounding. TMM 2023.

[2] CLIPREC: Graph-Based Domain Adaptive Network for Zero-Shot Referring Expression Comprehension. TMM 2023.

[3] Refclip: A universal teacher for weakly supervised referring expression comprehension. CVPR 2023.

[4] QueryMatch: A Query-based Contrastive Learning Framework for Weakly Supervised Visual Grounding. MM 2024.

[5] Part-Aware Prompt Tuning for Weakly Supervised Referring Expression Grounding. MMM 2024.

**Questions:**

See weakness.

**Details Of Ethics Concerns:**

No Ethics Concerns.

---

### Note · Authors · 2024-11-12

I have read and agree with the venue's withdrawal policy on behalf of myself and my co-authors.